# Evaluation of Marine Recreational Fisheries and Their Relation to Sustainability of Fisheries Resources in Greece

**Anastasios Papadopoulos** [1],*, **Konstantinos Touloumis** [1], **Emmanouil Tziolas** [1], **Dimitrios Boulamatsis** [2] **and Emmanouil Koutrakis** [1]

[1] Department of Marine Fisheries, Fisheries Research Institute (FRI), Hellenic Agricultural Organization "DIMITRA", Nea Peramos, 64007 Kavala, Greece; touloumisk@inale.gr (K.T.); tziolasm@inale.gr (E.T.); manosk@inale.gr (E.K.)

[2] Metron Analysis, Greek Market & Opinion Research Company, Sinopis 6, 11527 Athens, Greece; dboulamatsis@metronanalysis.gr

\* Correspondence: apapadop@inale.gr; Tel.: +30-2594022691

**Abstract:** During 2019, Greece conducted a nationwide telephone survey in order to estimate the number of marine recreational fishers (MRF), to identify the main fishing methods, to estimate their effort, economic aspects of their activity and to evaluate certain perceptions of marine recreational fishing. MRF are estimated to be 7.93% of the country's resident population. Three main fishing methods were identified, fishing from the shore, fishing by boat and spearfishing. It is estimated that Greek MRF carry out approximately 11,461,765 fishing trips/year, their total catch is 9350 t/year and the total yearly expenditure is 132,186,000 Euros. Marine recreational catches constitute one-third of the reported small scale coastal fisheries' catches, with their main catches belonging to the Sparidae family. The majority of MRFs are aware of the rules and limits associated with their activity, however a significant percentage seems to ignore or deny their existence. The results of the study indicate the social and economic importance of the activity in Greece and clearly demonstrates the need for regular monitoring and field work in order to properly evaluate marine recreational fisheries and sustainably manage both the activity and fisheries resources in the country.

**Keywords:** marine recreational fisheries; telephone surveys; fishing methods; demographics; Mediterranean; sustainability; Greece

## 1. Introduction

Marine recreational fisheries include all the relevant leisure and sport activities linked to the exploitation of marine living aquatic resources. Apart from the sport and leisure aspects, recreational fishing often satisfies subsistence purposes, a phenomenon that is increasing rapidly in recent years—especially in Southern Europe due to economic uncertainty [1], which complicates the classification between pleasure and subsistence purposes [2]. The difficulty of classifying subsistence fishing (commercial or recreational) is reported by Arlinghaus and Cooke [3], thus complicating the identification of socioeconomic and ecological benefits solely from recreational fisheries. Presumably the one perspective does not prove the other wrong, though the alternative definitions develop manifold motivations among marine recreational fishers (MRF) and consequently different managerial approaches for each country depending on socioeconomic criteria. Moreover, monitoring and assessing the impacts of recreational fisheries constitutes an even more complex managerial task, considering that there are many fishing modes with different characteristics.

Worldwide, fisheries data generally omit catches from recreational fisheries as well as catches from illegal-unreported fishing activities [4], partially due to the uncertainty and scarcity of relevant data (number of MRF and catches) [5,6]. On a global level, estimates of MRF vary widely from 220 million to 700 million, on average 10.52% of the total population

of the industrialized countries [7]. In Europe, a recent study [6] estimates 8.7 million European MRF, corresponding to a participation rate of 1.6% of the population with Greece having the highest (2.7%) participation rate (some 300,000 MRF). The study underlines that available data in the continent are limited or outdated despite national efforts to estimate MRF participation. Recreational catches have been estimated at ~800,000 t globally in 2010 [4]; however, authors have characterized their attempt on recreational caches as preliminary. Marine recreational catches account for slightly less than 1% of total global marine catches, with trends varying regionally [8]. Furthermore, additional necessary data, such as species individual weight, length and fishing effort for recreational fisheries are generally missing, a fact that impedes the assessment of fisheries catches and discards in the oceans [9,10]. Since the impact of marine recreational fisheries on fish stocks is far from negligible, the lack of relevant data does not allow the formation of cumulative management strategy for marine recourses that includes recreational fisheries, raising the uncertainty for the sustainability of fisheries resources and ecosystems.

In the Mediterranean Sea, management of marine resources is gaining increasingly more attention, since fisheries overexploitation imposes pressure to the natural ecosystems [11]. The necessity for updated information is evident today more than ever, while marine recreational fisheries data should be included along with commercial fisheries outputs, in order to create an integrated sustainable management approach for marine recourses. This notion is enhanced, since in specific occasions and regions, recreational catches could exceed commercial catches as in Cyprus [12] or be close to commercial catches, as in coastal areas of Spain for artisanal catches [13]. Furthermore, the competition for common resources between MRF and commercial fishers is enhanced by the activity of IUU fishers, such as retired professional fishers who illegally continue their professional activities, or MRF who likewise illegally sell their catches regularly. This highlights an additional significant management issue of the Greek fisheries policy. Recreational fisheries include all the relevant leisure and sport activities linked to the exploitation of marine living aquatic resources. Any type of commercial trading of recreational catches is prohibited [14], while commercial fisheries refer to the professional activity, aiming at economic benefits via trade [15].

In Greece recreational fishing has always been under the responsibility of the Ministry of Rural Development and Food while the control of the implementation of the relevant regulations was under the responsibility of the Hellenic Coast Guard which today is administratively subordinated to the Ministry of Maritime Affairs and Insular Policy. Historically the state's approach to the regulation of fisheries in Greece was much inspired by the principle that access to fisheries should be left quite open, an approach that has always been a source of tension between professional fishers and MRF. Recreational fishing is mentioned for the first time in a legal document, Royal Decree 666/1966, which distinguishes between professional and recreational licenses for fishing vessels. Then in the mid-eighties following a swift development of recreational fisheries and amidst strong reactions by the MRF and their organizations, Greece issued a Presidential Decree (373/85) as a first attempt to regulate the previously unmanaged activity, which in part and after amendments is still valid today. Apart from introducing some necessary rules, at the heart of this effort was the introduction of a licensing system which applied only to recreational fishing by boat. However, the process of issuing recreational fishing licenses has not been continuous and consistent [1]. Therefore, although Greece had for some time an indication of the number of MRF in the country, this was completely inaccurate since the vast majority of the MRF which fish from the shore have never been taken into account. Regardless, the licensing system was abolished in 2014, returning to an unmanaged regime, leaving again space for an unknown number of vessels and MRF (local and foreign), to engage in this activity. Nowadays, only professional fishers are legally required to have a license, while MRF do not, thus making anyone fishing without a license an MRF. Additionally, the absence of any census regarding the number of MRF (either Greeks or tourists) hinders the evaluation of the activity, while the extensive coastline and the high number of inhabited

and uninhabited islands, constitutes a complicated environment regarding the collection and processing of primary data on a regular basis.

Existing studies used various methods for estimating the actual number of MRF in Greece although the number of papers presenting results regarding fisheries resources are quite low [6,10]. Using official data from the formally existing licensing system [16] reports 96,075 registered MRF (which corresponds to 0.87% of the country's population if we arbitrarily use the 2001 census population [17] as a measure of comparison) noting that this value represent only boat fishing thus they are underestimations due to the fact that not all MRF applied for a license. Furthermore [5], using various assumptions 'conservatively assumed' the shore based MRF to be 1.5% of the coastal population. Their approximation is based on a conservative assumption of 1.5% and 1% of the coastal residents, which seems to be much greater in the present study.

The collection of available data on the number of MRF, as well as on various aspects of their activity at a national level, was the main goal of the pilot study regarding recreational fisheries in Greece that took place during the period 2018–2019 and has been extended for the period 2020–2021. The pilot study was conducted within the framework of National Data Collection Program of Fisheries Data (program co-funded by the EU through Fisheries and Maritime Operational Program 2014–2020) under the supervision the Ministry of Rural Development & Food, following the EU implementing decisions on management and use of data in the Fisheries and Aquaculture Sector [18,19]. These decisions were the cornerstone of recent Data Collection Framework programs (2017–2019 and 2020–2021). The final objective of the pilot studies, requested by all Member States, was the integration of the recreational catches sampling scheme in the regular sampling scheme of the multiannual Union program after 2022. As part of the Greek pilot study, a nation-wide telephone survey was conducted twice by the Fisheries Research Institute in collaboration with Metron Analysis, a market research and opinion polls agency. A preliminary survey with a sample of 5500 telephone interviews was conducted in 2018 in order to standardize the methodology and identify possible weaknesses. The main survey was conducted in 2019 aiming to increase precision and fill the gap of information on the engagement of the general population to recreational fishing, due to the absence of a registration process at national level.

The aim of the current study is to present the main outcomes of the 2019 nation-wide telephone survey, focusing on the estimation of the active number of MRF resident in Greece and their demographics, the identification of their main methods of fishing, the estimation of their avidity, catches and expenses, the specification of their awareness and social stance towards management of recreational fishery and examine possible relation to sustainability of fisheries resources in Greece. This effort, including all aspects of recreational fisheries, is done for the first time in a national scale in Greece.

## 2. Materials and Methods

The telephone survey conducted in the frame of the Greek multiannual Union program for the Data Collection in the Fisheries and Aquaculture sectors [20], between 18 April 2019 and 24 July 2019 using a sample of 16,501 telephone interviews. A structured questionnaire designed specifically for MRF was used, through Computer Assisted telephone interviews (CATI). Telephone interviews (19%) were checked by the method of co-listening, while the consistency of the questionnaires was checked electronically at 100%. The survey's main purposes were the estimation of the number of MRF in Greece and additionally the recording of their demographic characteristics such as sex, age and education level. As MRF was considered anyone older than 15 years of age engaged at least once in recreational fishing during the past 12 months prior to the survey.

The telephone interview unfolded in two phases: first a two-minute interview with an introduction for the survey and a screening question on whether the respondent is an MRF or not. If the respondent was not an MRF the interview was ended. If the respondent was

an MRF the interview continued with the questionnaire in order to obtain relevant answers as well as socio-demographic details.

The researched population was men and women, residents of the country who could communicate in Greek, regardless of nationality. The selection of the sample was based on a population consisted of 9,247,018 people, based on the most recent (2011) population census of Hellenic Statistical Authority [21]. The sampling frame of the survey emerged from randomly selected fixed and mobile phones (50:50 ratio) with quotas in terms of distribution in the basic administrative unit in Greece, namely the Regional Unit (RU).

The sample was spatially stratified in order to be distributed appropriately in all RUs and from all urban, semi-urban (over 2000 inhabitants) and rural areas (less than 2000 inhabitants) of mainland and islands. Furthermore, the sample was weighted, based on the RU's population distribution and the proximity of the RU to the sea using three categories: insular RU, RU's with access to the sea and landlocked or limited access to the sea RU's. Insular RU's and RU's with access to the sea were oversampled in order to increase the possibility of MRF participation. Additional weightings of the sample were also applied, based on the educational level and the distribution of gender by age groups, in order to address deviations from the population.

During the research, the sampling distributions based on gender, age and educational level of the respondents were monitored. Successful telephone interviews, denials, as well as absences which occurred during the telephone calls were also monitored. In case of absence, a second call was made at a different time and/or day as a second attempt to find the respondent. In case the second call was not answered, then this household was replaced by another. The households that agreed to participate in the survey fall into the category "successful telephone interviews". In cases where there was more than one member of the household engaged in recreational fishing, this information was also recorded but it was not included in the final estimation. In addition to the physical impact of recreational fishing the awareness and social stance of MRF under the current legislative framework has been investigated in this study as well.

The survey's questionnaire consisted of 20 numbered questions. In addition to demographics, most of them were closed questions, whilst other questions allowed respondents to answer spontaneously for issues such as modes of fishing (namely: (a) fishing from the shore, (b) fishing by boat, (c) spear fishing, (d) other), avidity (number of fishing trips per year), expenses (the total expenditure per fisher related to the activity during the past 12 months prior to the survey in euros spent per year). There were also some open-ended questions, such traveling to nonresident RU's for fishing by the respondent, the three most caught species of fish in terms of weight, and their perceptions and proposals regarding management of recreational fishing in the country. The average duration of interviews was about 5 min. During this process all prescribed rules of personal data protection were complied with, and all the collected data were anonymously analyzed.

The percentage of MRF per RU was estimated as the ratio of MRF to the sum of the participants to the survey. The estimation of the total MRF population (older than 15 years of age) has been done by post-stratifying by sex and raising to total population per RU, using the r package survey (function *svytotal*). Calculation of the studied variables (a) Age, (b) avidity, (c) Expenses, (d) Catches in kg/year in relation to the modes of fishing in Greece (shore fishing, by boat, spearfishing) was based on the MRF that fish solely with the corresponding mode of fishing. The sampling errors of the survey for a percentage of $p$, have been calculated with the assumption of simple random sampling at a confidence level of 95%, as: Sampling Error = $1.96 * \sqrt{(p * q/n)}$ where $p$ is the estimated percentage, $q = 1 - p$ and $n$ is the sample size. Statistical tests to identify statistically significant differences between socio-demographic categories for each question of the questionnaire were performed at a significance level of 0.01% ($p$ value < 0.01). The $\chi^2$ test was used to compare responses between discrete variables. One-way analysis of variance (ANOVA) was used to compare discrete and quantitative variables and R Studio version 4.0.3 [22] was used for further analysis and presentation.

## 3. Results

A total of 16,501 telephone interviews were conducted during the telephone survey of 2019. The overall response rate was 17% which corresponds to the ratio of interviews to total interviews and denials. Of the 16,501 interviews, 1138 were MRF, of whom 1104 were above 15 years of age.

The total number of MRF in Greece was estimated to be 730,514 (SE = 0.004). This number corresponds to 7.93% of the total population of the country (within the ages of 15–89 years old). Considering the spatial distribution of MRF (Figure 1), insular areas and areas of the mainland with access to the sea, have a significant number of MRF. Notably, the islands of the Aegean depict a strong relationship ($p < 0.01$) with recreational fishing in comparison to the western region of the country and areas that have minimum or no access to sea which depict lower numbers. Large urban centers such Attica show 8% and Thessaloniki 10% participation rate, respectively.

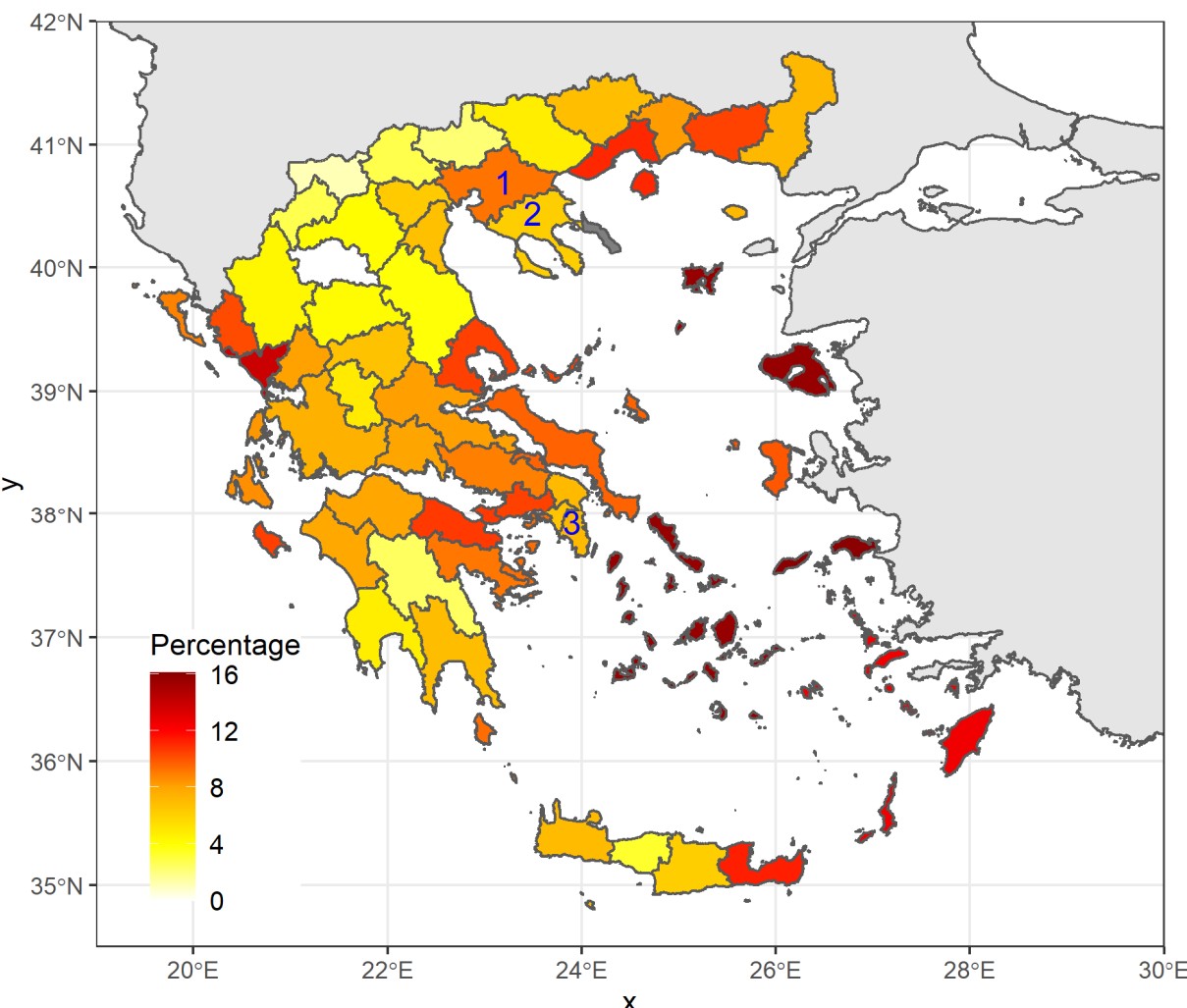

**Figure 1.** Spatial distribution of MRF (percentage of the total population) in the Greek Regional Units. (**1**) Thessaloniki, (**2**) Chalkidiki, (**3**) Attica. Mount Athos (dark gray area) is an autonomous part of the Greek State and was not included in this study.

In terms of households, 16% (~650,000) are estimated to have at least one member engaged in recreational fishing. Participation appears enhanced within large households ($p < 0.01$) where participation rate is higher (18%) in those households with more than four members in comparison to single member households (13%). Two and three member households show 15% participation rate. Households in both North and South Aegean

depict higher participation rate with three out of ten having at least one member who practices recreational fishing.

Participation in the general population reaches 12.52% (SE = 0.005) of men versus 3.66% (SE = 0.003) of women. Those aged 25–34 years old seem to constitute the most active MRF in both sexes declining in numbers as the age increases (Figure 2).

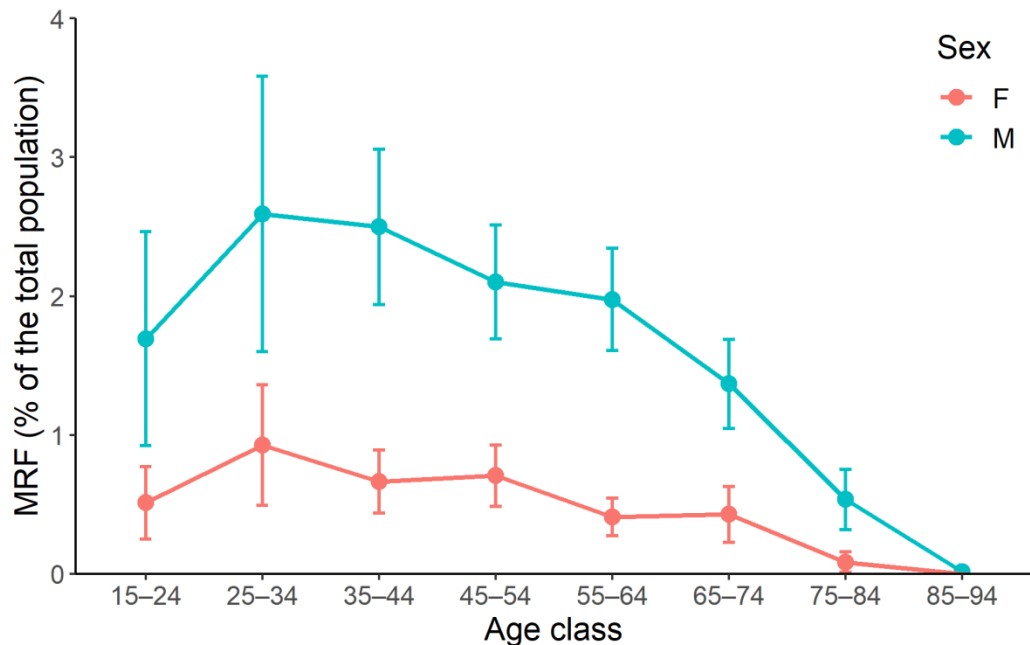

**Figure 2.** Ratio between sexes and age groups of MRF in Greece. Error bars represent 95% Confidence Intervals of the estimations.

In terms of age per mode of fishing (Figure 3) boat MRF's have an average age of 47.85 years (median 48.00, SE = 0.74), shore fishing MRF's 42.59 years (median 43.00, SE = 0.62), and spearfishers 41.05 years (median 41.00, SE = 1.08). The average age of the Greek MRF is 49.40 years old (median 49.00, SE = 0.148).

The main fishing pressure, in terms of number of MRF fishing at RU level annually, is situated in Attica (Athens) and its surrounding RUs and Chalkidiki and its surrounding RUs (Figure 4). The remaining coastal areas of the mainland as well as the islands receive moderate fishing pressure, whilst naturally areas with no access to the sea illustrate no fishing pressure. Analysis of travelling pattern show that 60.80% (SE = 0.028) of MRF choose locations situated within their RU of residence especially in the islands (95%, $p < 0.01$%). Most of the MRF that remain within their residence RU belong to the age group of 55–64 years old, have lower education and usually they are freelance or employees of the public sector. Conversely, 28.96% (SE = 0.026) of the MRF travel to neighboring RU, especially MRF from landlocked RUs (46%, $p < 0.01$%) as well as from the RU of Thessaloniki (77%, $p < 0.01$%). Furthermore 30.80% (SE = 0.027) of MRF travel further distances for fishing (neither to home RU nor to a neighboring RU), mainly from Attica (54% $p < 0.01$%). Traveling for fishing in non neighboring RU's is associated with higher education and small size households.

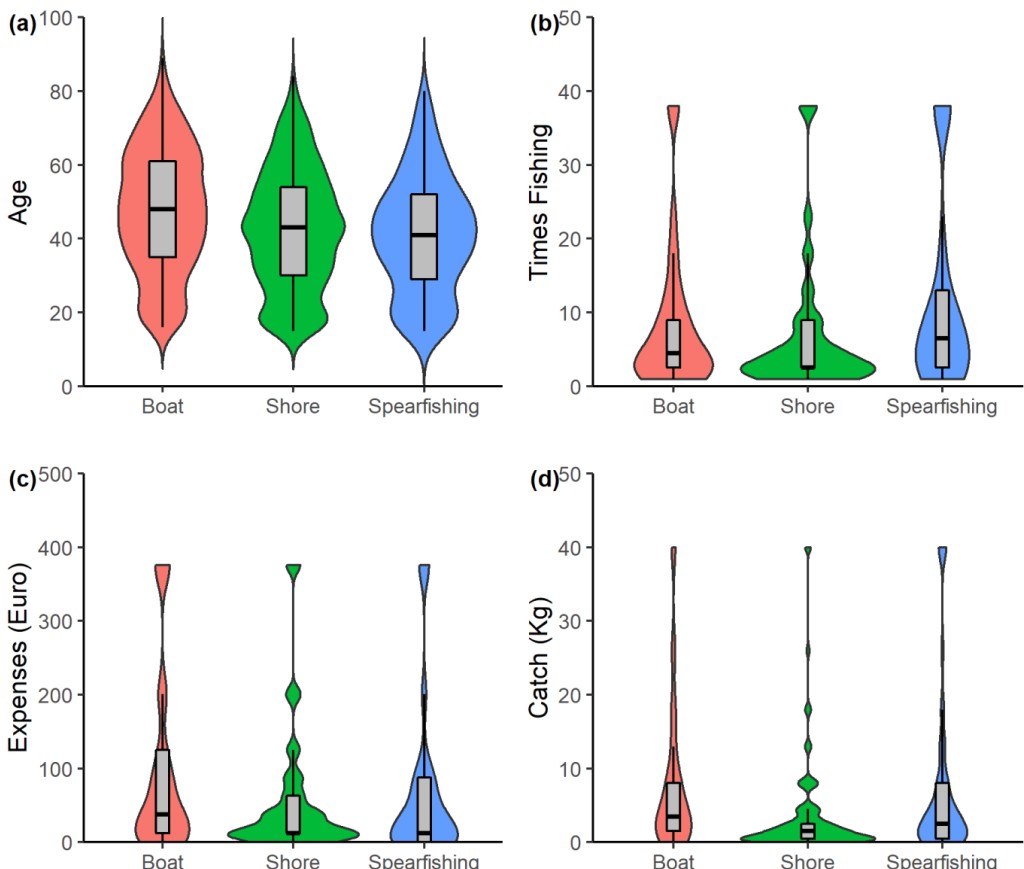

**Figure 3.** Box plots of the studied variables (**a**) Age, (**b**) Times of fishing, (**c**) Expenses, (**d**) Catches (kg/year) in relation to the modes of fishing in Greece (boat, shore, spearfishing). The colored shapes around the boxplots represent the distribution of the data around different values per fishing mode.

There are three fishing methods (modes) that are most common in the country (Figure 5). More specifically, at least once per year 44.62% (SE = 0.029) of MRF are fishing solely from the coast, 21.88% (SE = 0.024) are fishing exclusively by boat and a subgroup of 10.70% (SE = 0.018) is fishing by both modes. Only 8.86% (SE = 0.017) are exclusively spearfishers, but together with the overlapping subgroups spearfishing and boat 4.24% (SE = 0.001), spearfishing and coast 3.99% (SE = 0.011), all types of fishing 3.50% (SE = 0.01), they exceed 20% of the MRF in the country. Nevertheless, spearfishing is considered as a third pole of MRF, since the aggregate sum of MRF by boat and from the shore is exceeding 65% of the total MRF. A small percentage (2.21%) is using other methods of fishing.

The use of line/fishing rod is higher than any other method of fishing (62.80%, SE = 2.80) from the coast. It is a very common method, used especially by the unemployed (81%, $p < 0.01$), women (71%, $p < 0.01$), those aged 55–64 years old (71%, $p < 0.01$) and MRF coming from areas with restricted or no access to the sea (77%, $p < 0.01$). It is also the more commonly used method in the north parts of Greece, namely East Macedonia and Thrace (76%).

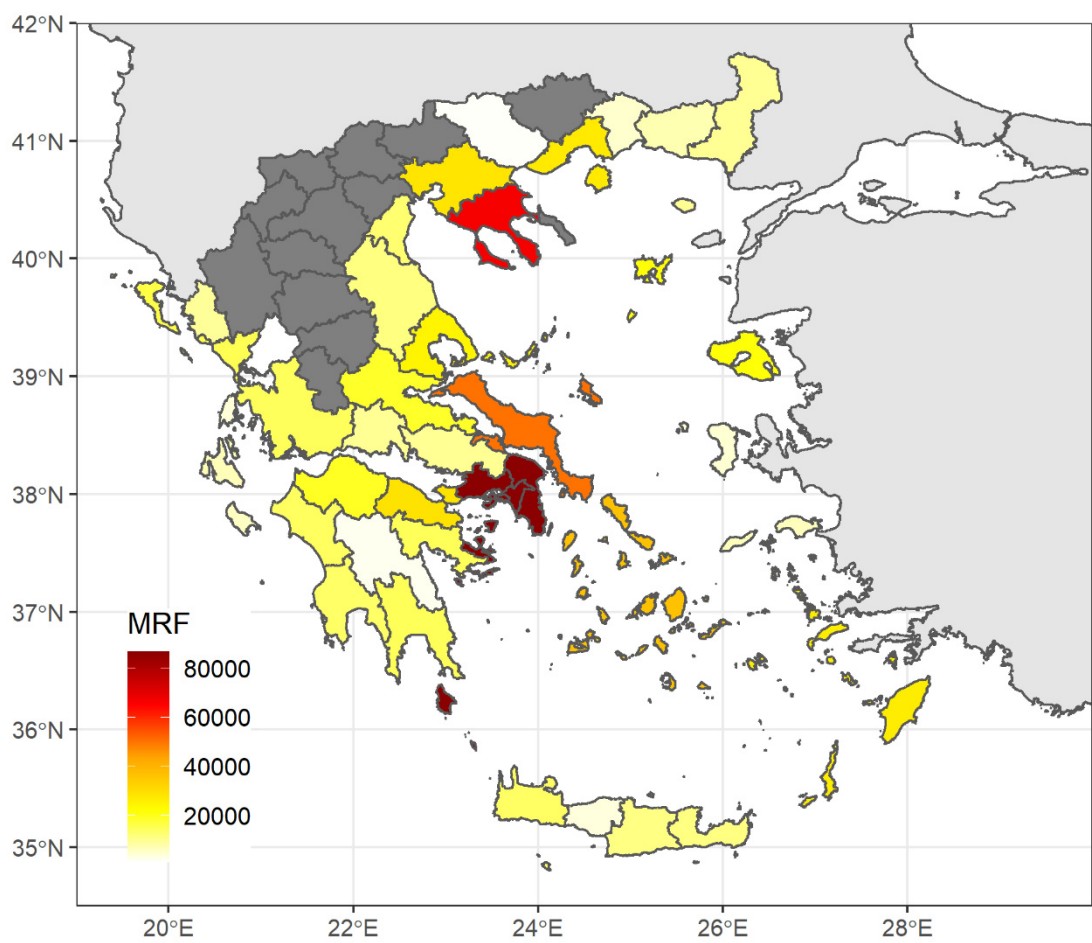

**Figure 4.** Fishing pressure in terms of the estimated total number of MRF operating annually at each Regional Unit of Greece (areas in dark grey are RUs with zero fishing pressure).

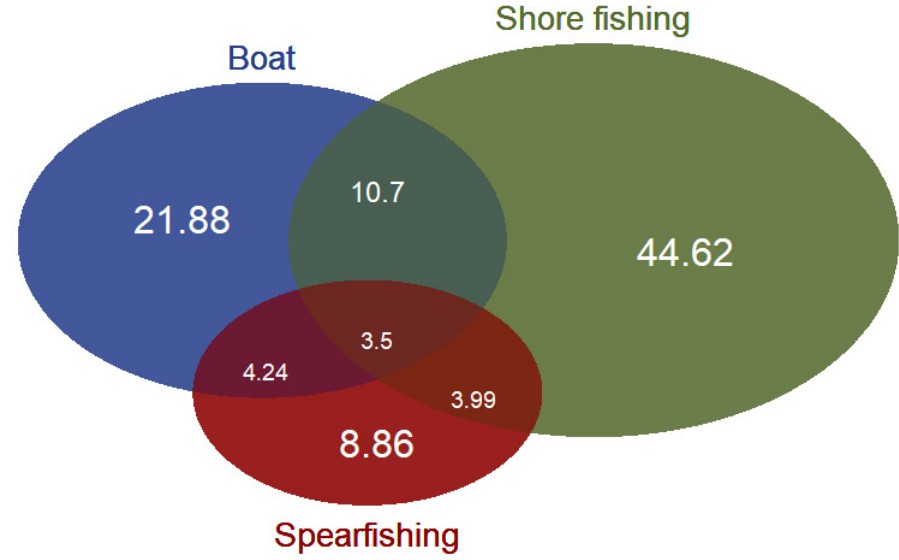

**Figure 5.** Percentage of MRF participation per mode of fishing in Greece and relevant overlapping subgroups.

The use of line and fishing rod from boat 37.27 (SE = 0.028) is higher in younger ages (50%, $p < 0.01$) compared to retirees (49%, $p < 0.01$) and individuals of higher education (45%,

$p < 0.01$). Boat fishing is practiced by virtually all demographic groups with an indication ($p < 0.01$) that there is higher participation by retirees and younger ages associated with higher education. Fishing by boat using longline is more common in Attica (15%) and the islands (13%). Spearfishing is more common amongst men (24%) of younger age 15–24 years old (24%) and higher education (26%).

In terms of avidity more than half (52.51% of the MRF, SE = 0.029) are occasional MRF, fishing 1–5 times per year, 17.64% (SE = 0.022) up to 6–10 times, 14.06% (SE = 0.02) up to 11–25 times, 8.62% (SE = 0.016) are more motivated MRF, fishing up to 26–50 times and 7% are really fishing enthusiasts with more than 50 times per year. Seen by mode of fishing (Figure 3) boat fishers fish on average 12.36 times/year (median 4.50, SE = 1.21), shore fishers 9.55 times/year (median 3.62, SE = 0.75) and spearfishers 16.45 times/year (median 9.00, SE = 2.25). Statistical analysis ($p < 0.01$) show that men belong to the 55–64 age group are more avid MRF (26%). Higher avidity is associated with retirees (24 times/year), married men (17 times/year) or singles with partners (19 times/year) than singles living with their parents. Geographically, MRF living on islands are significantly more avid (20 times/year) than others.

The average times of fishing for all types of fishing is 15.69 (SE = 0.65) though the median value is seven times per year. The difference is considered important, though it is explained by the frequent visits of regular MRFs identified in Figure 3b.

Annual declared catches show that the vast majority of MRF (65.52%) catch small amounts of fish, up to 5 kg/y, 15% up to 15 kg/year and 8% up to 30 kg/year. A small minority of 8% within the MRF community are fishing up to 30 kg/year and 9.97% more than 50 kg/year. In terms of mode of fishing (Figure 3), spearfishers with 13.44 kg/year on average (median 3, SE = 2.57) and boat fishers with 13.16 kg/year on average (median 4.5, SE = 1.42) depict similar efficiency. Shore fishers seem to catch far less from the other two modes with 4.40 kg/year on average (median 1.50, SE = 0.53). Statistical analysis shows that men of the 55–64 age group (21%) and retirees (20%) land more catches and clearly the most effective MRF live in the islands (23%) and in the Peloponnese region unit (26%).

On average an MRF catch is 12.80 kg/year (SE = 1.10); this adds up to 9350.6 t/year. As shown in Figure 6 this amounts to almost one third (31.6%) of the small scale fishery landings (29,360.5 t), 17.7% of the open sea fishery landings (52,559.5 t) and 11.35% compared to the total commercial fleet landings (81,920 t), the same year of the study [23]).

A total of 53 taxa have been identified (Figure 7) as a response to the question of three MRF most caught species in terms of weight in the last 12 months. The Sparidae family dominates the recreational catches in the Greek waters with the top three of the most caught species: gilthead seabream (*Sparus aurata*) is the most commonly caught species (1 in 4 MRF caught species) followed by annular seabream (*Diplodus annularis*), and white seabream (*Diplodus sargus*). These three species together with seven additional Sparidae species, namely striped seabream (*Lithognathus mormyrus)*, common pandora (*Pagellus erythrinus*), bogue (*Boops boops*), blackspot seabream (*Pagellus bogaraveo*), saddled seabream (*Oblada melanura*), Mediterranean parrotfish (*Sparisoma cretense*) and red porgy (*Pagrus pagrus*), comprise more than half of the main 19 species caught. Geographical analysis revealed significand ($p < 0.01$) differences in the catches. In areas with access to sea, *S. aurata*, *L. mormyrus* and *P. erythrinus* are more common catches, while in insular areas *D. sargus*, *S. credence* and *Epinephelus marginatus* constitute the main species caught. *Mugil cephalus* prevails in catches in areas with limited access to the sea.

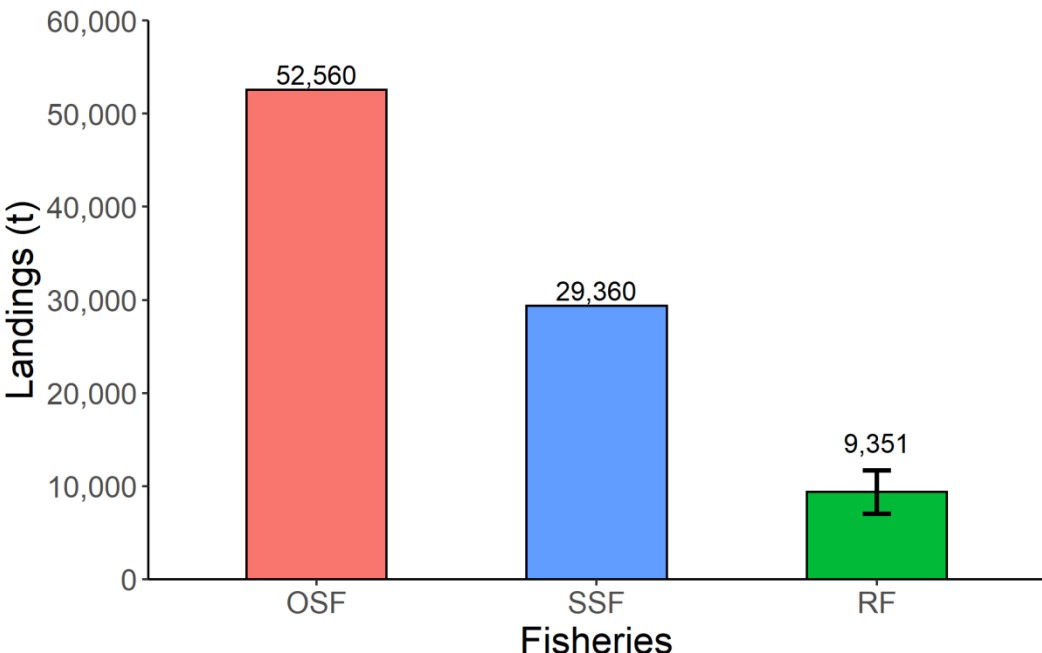

**Figure 6.** Estimated MRF catches compared to commercial fishing fleet catches (OSF = Open Sea Fisheries, SSF = Small Scale Fisheries, RF = Recreational fisheries).

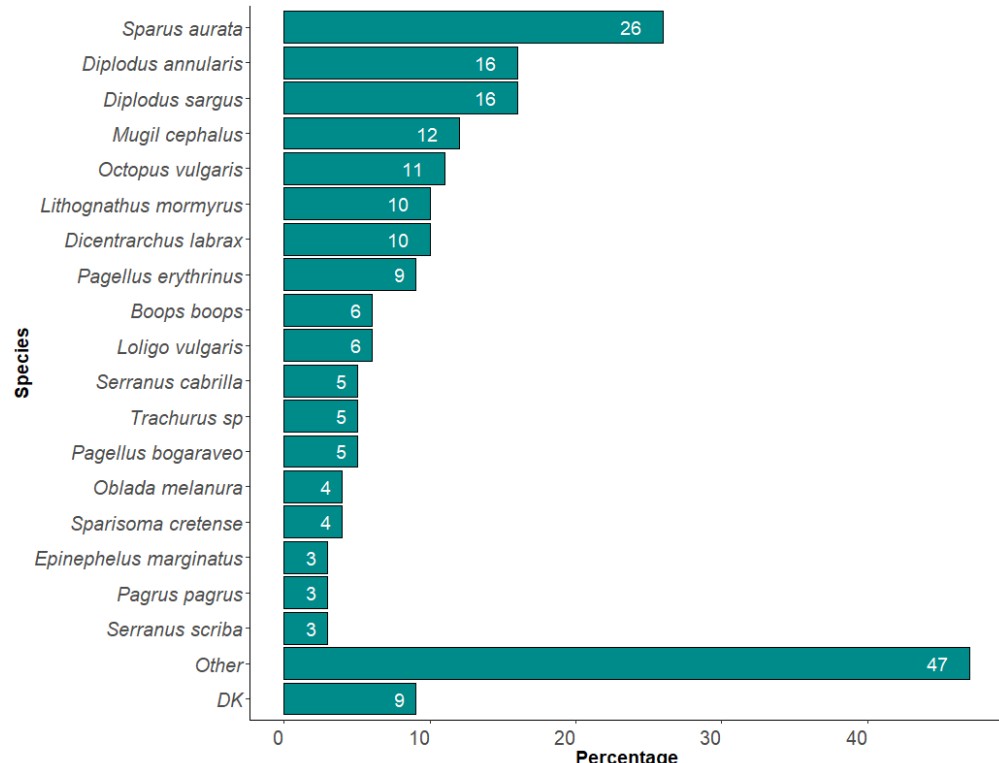

**Figure 7.** Reported caught species (% share of catches in weight) in Greece, for the study period (DK: Don't Know).

Regarding annual expenses a large portion 13.25% (SE = 0.02) of MRF reported zero expenses for recreational fishing in the previous 12-month period. However, the large majority 43.75% (SE = 0.029) of MRF spent small amounts of money for their activity, between 1 and 50 €/year, while 13.81% (SE = 0.02) and 11.64% (SE = 0.02) of MRF de-

clared expenses of 51–100 €/year and 101–250 €/year, respectively. There is, however, a portion of 7.48% (SE = 0.01) of the MRF community that reported relatively large expenses 251–500 €/year with enthusiasts 8.84% (SE = 0.016) reporting sums of 501 €/year up to thousands of Euros per year spent for recreational fishing. Apparently, (Figure 3) MRF from the coast spent less on average 79.08 €/year (median 12.50, SE = 8.41) in comparison to boat 203.78 €/year (median 37.50, SE = 23.17), whilst spearfishers spent 155.93 €/year (median 37.5, SE = 31.83). The average expenses for all types of fishing are 180.95 €/year (median 38.0 €/year, SE = 14.44) raising the total amount of money spent on recreational fishing to approximately 132,186,508 €/year. Statistically significant differences ($p < 0.01$) are related to avidity, sex, age, education and employment. Men between 55–64 years of age of higher education and usually freelancers are more avid and spend larger amounts of money, whilst low spending is associated with unemployed individuals and housewives. There are no significant differences related to family status or geographical analysis.

Awareness and social stance of MRF under the current legislative framework, are presented in Figure 8a.

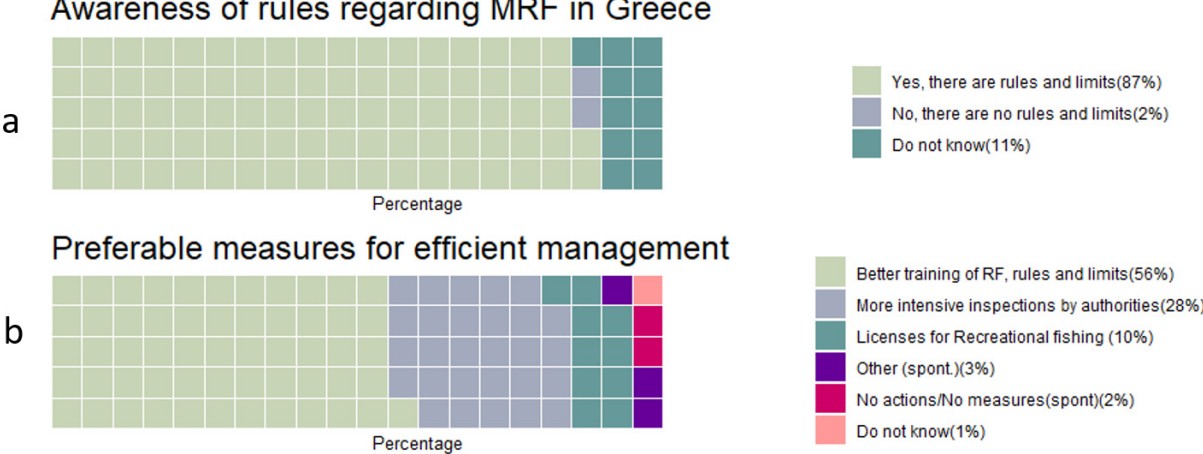

**Figure 8.** Waffle charts: (**a**) Percentage of awareness regarding rules and limits for the Greek recreational fisheries (**b**) MRF's suggested measure rate, for efficient management of recreational fishing in Greece.

The vast majority of MRF (86.97%, SE = 0.02) know that there are rules and limits in recreational fishing, but a significant percentage (11.04%, SE = −0.02) do not know if there are rules or limits, whilst 2.00% (SE = 0.01) say there are no rules and limits. Overall, it is estimated that more than 90,000 MRF approximately (13.4%), are not aware about the rules and limits governing the activity they are partaking in. Awareness regarding rules and limits is high in all demographic groups and across the country, showing no differentiation based on geographical analysis. Awareness is lower ($p < 0.01$) amongst younger MRF between 16–24 years of age, unemployed and housewives while those aged between of 35–44 and employees of the private sector are more aware.

MRF were also asked to suggest preferable measures for the more efficient management of MRF in the country (Figure 8b). According to their views, 55.68% (SE = 0.03) suggested that the best way for efficient management is training/information regarding rules and limits, 27.63% (SE = 0.03) urge for more intensive control by the authorities, 10.12% (SE = 0.02) believes that issuing of licenses is the best way forward, 2.08% (SE = 0.01) claims that there is no need for any measure, 3.44% (SE = 0.01) suggests other measures and 1.06% (SE = 0.01) has not expressed any suggestions. There is high uniformity in all demographic groups and across the country showing no differentiation in preferences based on geographical analysis. However significant differences ($p < 0.01$) can be detected amongst MRF aged between 25–34 years and unemployed who are in favor of better training, while married MRF are in favor of stricter controls.

## 4. Discussion

Greece abolished in 2014 an established albeit dysfunctional structure for monitoring marine recreational fishing. That consequently resulted in an incalculable number of MRF, making the assessment difficult and the sustainable management of marine recreational fishing activity even more difficult. Simultaneously, studies attempted to assess recreational fisheries were scarce and fragmentary.

In an effort to overcome this uncertainty, the current work followed a holistic approach, covered all modes of fishing in the whole country, presenting results that can be used as the basis for future management policies.

According to the results of the present study some 325,000 (44.62%) fish solely from the shore. That number is raised to some 63%, if shore fishing MRF fishing with other modes as well are included. Fishing from the shore is a mode of fishing for which the participants were not obligated to apply for a license neither have they ever been monitored or assessed in the past. They constitute the vast majority of the MRF community in the country, increasing their number to more realistic levels and fall close to the 68% reported in the area of East Mediterranean by [24]. This number increases further if foreign MRF are added to the Greek MRF, which according to a study under preparation constitute about 45% of the total number of MRF in the northern provinces of the country (i.e., East Macedonia and Thrace), bringing forward additional concerns for sustainable management.

Although in Greece marine recreational fishing is evidently a male activity, participation rate of women and especially women of younger age in MRF seems to be high in the country, compared to the industrialized countries [7] and most parts of the Mediterranean, where the activity is heavily dominated by men [12,24–27], but this result could also be a biased approach of the survey and needs further investigation. An interesting point revealed is that women mainly use lines from the shore, a less effective method of fishing.

Regarding age the present study shows that there is a relationship between age and mode of fishing and should be further investigated. For example, boat fishing, which implies considerable higher expenditure but also higher catches, is more common in older ages a groups that seem to be able to afford the cost. Spearfishing a demanding more selective mode of fishing that can affect the abundance and reproductive cycle of long lived, slow growing, top predator species such as the groupers [28–30] is exercised by younger ages on average which in general are in better physical condition. Shore fishing in turn is practiced by all ages probably due to ease of access, lower associated costs and less specialized skills.

Travelling for fishing seems to be associated with the proximity to the sea. The lower frequency of movement is in the islands, where obviously there is no need to dedicate much time, travel or money to fish but travelling from the two large urban centers of Attica (Athens) and Thessaloniki is also high imposing increased pressure to the fished species and their ecosystem in nearby areas due to a larger number of MRF.

Although the number of MRF is high in Greece, it seems that Greeks are not avid MRF. The average days of fishing per fisher for all modes is small with the vast majority of MRF fishing few times per year. This value varies greatly from those reported for the country from [16] 60–90 days/year, [5] 180–193 days/year and [10], 81–104 days/year, showing large deviations between them and the present study as well. Comparing these estimations with the rest of Europe, [6] reports 6.79 days/year per MRF in Greece and on average 9 days/year per MRF in Europe, a value closer to the one reported in the present study. It is worth noting however that recreational fishing effort in the Mediterranean is underestimated.

Regarding catches, the Sparidae family dominates MRF catches in the Greek waters following the global trend where recent catches were dominated by Sparidae (12% of total catches) [8]. The substantial share of gilthead seabream (*Sparus aurata*) in Greek recreational catches is noticeable and is also reported in other related studies in the Greek waters [5,10], and also in Cyprus [12]. The species abundance is a common feature in Mediterranean waters and can be largely attributed to fish farming. Gilthead seabream and European

seabass (*Dicentrarchus labrax*) are the main reared species of the 302 sea farms [31] in Greece which are scattered in high densities in both the Central Aegean and Ionian Sea as well as in the Aegean coasts of Turkey where sea bass and sea bream farms are localized mainly along the Aegean [32].

It has been reported that big schools of tens or hundreds of thousands of fish and several tons in biomass escapees from fish farms in Cyprus [33], contribute substantially to the MRF catches [12]. Increases in the local population of gilthead seabream in three areas along the southeastern Adriatic Sea are also attributed to escapes from local fish farm which enrich the local population constantly with new fish [34]. Increases in the number of local population was also attributed to the spawning of large mature fish inside fish farms in Messolonghi lagoons [35]. Furthermore, 'escape through spawning' in two farms in the Aegean and three farms in Ionian sea resulted to 351,000 to 702,000 fish per year that could recruit into wild populations [36]. These escapes increase the wild gilthead seabream populations which inhabits mainly coastal waters and estuaries where they first recruit and spread along the coast and islands at later stage, leading therefore to substantial increases in the overall population of gilthead seabream and consequent abundance of the species. In the case of European seabass (*Dicentrarchus labrax*) [33] suggests that long-term survival and interbreeding with wild fish is possible. The escape of fish from fish farms is considered as a threat to natural biodiversity in marine waters [37] with implications and risks of genetic and ecological importance, but regarding recreational fishing this can have both a significant socioeconomic impact by increasing MRF effort due to increased fishing chances [38] and consequently sustainability implications for other more vulnerable species.

Expenditure is low in the present study (€180.95) compared to the value of €680 that the average European MRF is spending annually [6]. Nevertheless, marine recreational fishing contributes more than €132 million to the national economy with many different economic sectors obtaining economic benefits from it.

Awareness regarding rules and limits in recreational fishing in the present study revealed that although the vast majority is aware that there are rules and limits in place, a considerable percentage of MRF states that they are unaware or appear certain that there are no rules and limits. This is estimated to be 1 out of 8 MRF. Lack of awareness is greater in the younger generations involved in MRF. This may be partly attributed to the abolition of the licensing system in 2014, through which there was an information channel about the existing rules and restrictions. Today the flux of information and clarity of communication is incomplete. The lack of communication between MRF and management authorities creates a significant information gap with implications for both MRF management and the sustainable management of fisheries resources and needs to be addressed. In addition, lack of communication certainly offers an excuse for the possible lack of awareness for foreign MRF, increasing furthermore the number of unaware MRF and their impact to the marine environment. It is promising, at the same time, that younger generations who are less aware of the rules and limits governing the activity find MRF training the way forward to better management of recreational fishing, a conspicuous signal to the managing authorities.

## 5. Conclusions

In summary, the present study showed aspects of recreational fishing that make the activity socially and economically important. The removal of a considerable amount of biomass has also biological implications. The sustainable management of recreational fishing in Greece is a difficult task for the country's authorities and requires planning and implementing a comprehensive management strategy in order to be successful. A combination of a licensing system that includes all marine recreational fishing modes, along with an effective control system, as well as training for MRF, seems to be of great importance for an overall successful sustainable management of marine recreational fishing. That combination will allow the competent authorities to achieve efficiency but also to provide valid estimates that can be used in stock assessment in the future. Concluding, the validity of the telephone survey will be put into perspective in future studies, since

the EU-MAP 2022–2024 promotes a holistic framework for the collection of credible primary data regarding recreational fisheries. On-site, face-to-face sampling will serve as a complimentary method of validating the results in various aspects of recreational fishing addressed in this paper, such as the possible misidentification of species and under or overestimation of catches, thus providing a more reliable representation of the composition of targeted taxa by MRF in Greece.

**Author Contributions:** Conceptualization, methodology, formal analysis, data curation writing—review & editing, A.P.; data analysis, visualization, review & editing, K.T.; formal analysis, data analysis, data curation, writing—review & editing, visualization, E.T.; formal analysis, data analysis, review & editing, D.B.; conceptualization, methodology, formal analysis, review & editing, E.K. All authors have read and agreed to the published version of the manuscript.

**Funding:** This research received no external funding.

**Institutional Review Board Statement:** Not Applicable.

**Informed Consent Statement:** Not Applicable.

**Data Availability Statement:** Not Applicable.

**Acknowledgments:** The authors would like to thank the Greek Ministry of Agriculture and Food which is responsible for the multiannual Union program for the Data Collection in the Fisheries and Aquaculture sectors (EU MAP).

**Conflicts of Interest:** The authors declare no conflict of interest.

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
