# Peer review of "Evaluation of Marine Recreational Fisheries and Their Relation to Sustainability of Fisheries Resources in Greece"

_sustainability, doi:10.3390/su14073824_

Round 1
Reviewer 1 Report
This study presents the main outcomes of the 2019 nation-wide telephone survey, focusing on the estimation of the active number of MRF resident in Greece and their demographics, the identification of their main methods of fishing, the estimation of their avidity, catches and expenses and the specification of their awareness and social stance towards management of recreational fishery. It presents the survey results well.
Specific comments:
- There are extra empty spaces. For example, line 120, ...conducted in the ....
- The 1st paragraph in the Discussion may be more appropriate for the Introduction.
Author Response
Dear sir/madam
Thank you for carefully reading our manuscript and offering your comments to improve it. We have taken them into consideration in preparing the revised form of manuscript. Below is our point-by-point response to your comments. The appropriate corrections to the revised manuscript are presented with track changes as suggested. We hope that our interventions based on your comments increased the quality of our work and render our manuscript suitable for publication.
Yours Sincerely
Anastasios Papadopoulos
‘This study presents the main outcomes of the 2019 nation-wide telephone survey, focusing on the estimation of the active number of MRF resident in Greece and their demographics, the identification of their main methods of fishing, the estimation of their avidity, catches and expenses and the specification of their awareness and social stance towards management of recreational fishery. It presents the survey results well’.
1) |
There are extra empty spaces. For example, line 120, ...conducted in the .... |
Reply: |
Several empty spaces are deleted throughout the whole manuscript. |
2) |
The 1st paragraph in the Discussion may be more appropriate for the Introduction. |
Reply: |
The 1st paragraph in the discussion was deleted. |
Reviewer 2 Report
This submission provides information from a survey of marine recreational fishers in Greece, for which there is limited data to date. The work is well designed and implemented with an adequate sample size to make national extrapolations. The manuscript fills a research need on MRF and provides good data and analysis of the frequency, demography etc. of fishers. The manuscript is worthy of publishing and contributing to the literature on recreation and sustainable fisheries. Very minor changes are proposed.
Main comments:
Line 184 – does the RU weighting mean that a 7.93% figure is confounded? Maybe you can also provide such a figure for each RU type.
Figure 2 – note in figure title what the error bars are – I assume SD, as the SEs quoted on the figures in text are very small due to large n
Line 203 – you say 25 – 35 is most active MRF, but in line 207 – 209 you say that all fishing modes have 49 years – is that due to demographics or why? Explain in text what this difference.
Figure 3 – the figure title says that they are box plots which is fine – but you also have the coloured blobs which I guess delimit the data – but you also need to say what these are in the figure title.
Figure 4 – Turkey etc are grey in figure as well – so be clear that the dark grey RUS’s in Greece are in grey
Line 234 and Fig 3 and Figure 5 – let’s just be consistent and say….from the shore. “Coastal” as used in Fig 3 and Fig 5 is confusing and it could mean fishing in coastal areas and not just the shore. So be more consistent and use shore fishing or fishing form the shore for this group – use “shore” as the name in graphs.
Line 271 – it might be worth saying that this is due to some very keen fishers in all groups as identified in Fig. 3b.
Minor changes:
Line 33 – full stop missing and reduce space between “..trend challenges…”
Line 34 – just explain what this link is in a few words.
Line 44 – just be clear here that no national reporting requirements for Greek or other EU member states.
Line 65 / 67 – you suggest this is a “conflict” – but it may be more appropriate to consider it as “competition for common resources” or similar – the use of conflict suggests some physical negative interactions between parties and as you talk of catch sizes I think it is more of a resource issue.
Line 90 – would be useful to define what is nationally recognised as recreational fisher as opposed to a commercial fisher (for small scale artisanal fishing what determines the difference?).
Line 104 – remove unnecessary space “..Framework programmes…”
Line 121 – check spaces, and line 362 and line 433
Line 203 – add space, and also Line 290
Line 281 – “catch” would be better than “fish”
Line 293 – caught not cought
Fig 7 – just be clear this percentage is by weight
Line 207 – expenses
Line 350 – delete?
Line 357 – dysfunctional
Line 395 – fishing
Line 519 – add in the EU finding under which this was implemented.
Author Response
Dear sir/madam
Thank you for carefully reading our manuscript and offering your comments to improve it. We have taken them into consideration in preparing the revised form of manuscript. Below is our point-by-point response to your comments. The appropriate corrections to the revised manuscript are presented with track changes as suggested. We hope that our interventions based on your comments increased the quality of our work and render our manuscript suitable for publication.
Yours Sincerely
Anastasios Papadopoulos
Comments and suggestions for authors
‘This submission provides information from a survey of marine recreational fishers in Greece, for which there is limited data to date. The work is well designed and implemented with an adequate sample size to make national extrapolations. The manuscript fills a research need on MRF and provides good data and analysis of the frequency, demography etc. of fishers. The manuscript is worthy of publishing and contributing to the literature on recreation and sustainable fisheries. Very minor changes are proposed.’
Main comments:
|
|
1) |
Line 184 – does the RU weighting mean that a 7.93% figure is confounded? Maybe you can also provide such a figure for each RU type. |
Reply: |
One of the main purposes of the research was to accurately estimate the number of recreational fishermen in Greece and therefore their percentage on the total population at a national level. Information on the percentage of Marine Recreational Fishermen per RU is provided through the Figure 1. |
2) |
Figure 2 – note in figure title what the error bars are – I assume SD, as the SEs quoted on the figures in text are very small due to large n |
Reply: |
The error bars represent the 95% confidence interval for the estimations. A sentence explaining that has been added to the Figure 2 legend. |
3) |
Line 203 – you say 25 – 35 is most active MRF, but in line 207 – 209 you say that all fishing modes have 49 years – is that due to demographics or why? Explain in text what this difference. |
Reply: |
This is indeed due to demographics. In figure 2 the percentage of MRF per age class is depicted. However (unfortunately) the age distribution in Greece is such that the age classes between ~40 to ~70 years old are the most populated. So, although the percentage of MRFs is higher in 25 – 35 years age class than in 45-55 years age class, the latter corresponds to more people (MRFs) than the former. |
4) |
Figure 3 – the figure title says that they are box plots which is fine – but you also have the coloured blobs which I guess delimit the data – but you also need to say what these are in the figure title. |
Reply: |
The “violin” plots (colored shapes around the boxplots) visualize the probability density of the sample at different values. We believe that this visual representation promptly conveys useful information on data distribution to the user. However, we acknowledge that some additional information on these plots should be added to the figure 3 legend. Please, see figure 3 legend. |
5) |
Figure 4 – Turkey etc are grey in figure as well – so be clear that the dark grey RUS’s in Greece are in grey. |
Reply: |
Legend of figure 4 is rephrased. |
6) |
Line 234 and Fig 3 and Figure 5 – let’s just be consistent and say….from the shore. “Coastal” as used in Fig 3 and Fig 5 is confusing and it could mean fishing in coastal areas and not just the shore. So be more consistent and use shore fishing or fishing form the shore for this group – use “shore” as the name in graphs. |
Reply: |
We agree, several changes have been done all over the manuscript, regarding the coastal and shore fishing mix up. Coastal is used only for describing an area and figures 3 and 5 have changed as well, in order to align with the comments. Corrected in the legend of figure 3. |
7) |
Line 271 – it might be worth saying that this is due to some very keen fishers in all groups as identified in Fig. 3b. |
Reply: |
We agree, it is noted and an explanation is added after the above sentence |
Minor changes:
|
|
1) |
Line 33 – full stop missing and reduce space between “..trend challenges…” |
Reply: |
Corrected |
2) |
Line 34 – just explain what this link is in a few words. |
Reply: |
The sentence is rephrased. (L. 33-37) |
3) |
Line 44 – just be clear here that no national reporting requirements for Greek or other EU member states. |
Reply: |
That was the case during the period of the study of Hyder et al 2018, [6] in reference list, but this study was not about EU countries only. For example non EU countries such as Albania and Montenegro were also included. EU countries today are required to report only total weight on recreational catches starting from 2022. |
4) |
Line 65 / 67 – you suggest this is a “conflict” – but it may be more appropriate to consider it as “competition for common resources” or similar – the use of conflict suggests some physical negative interactions between parties and as you talk of catch sizes I think it is more of a resource issue. |
Reply: |
Rephrased |
5) |
Line 90 – would be useful to define what is nationally recognised as recreational fisher as opposed to a commercial fisher (for small scale artisanal fishing what determines the difference?). |
Reply: |
A sentence was added specifying the difference between professional and MRF. |
6) |
Line 104 – remove unnecessary space “..Framework programmes…” |
Reply: |
Corrected |
7) |
Line 121 – check spaces, and line 362 and line 433 |
Reply: |
Corrected |
8) |
Line 203 – add space, and also Line 290 |
Reply: |
Corrected |
9) |
Line 281 – “catch” would be better than “fish” |
Reply: |
Corrected |
10) |
Line 293 – caught not cought |
Reply: |
Corrected |
11) |
Fig 7 – just be clear this percentage is by weight |
Reply: |
The legend is corrected |
12) |
Line 207 – expenses |
Reply: |
Corrected |
13) |
Line 350 – delete? |
Reply: |
Corrected |
14) |
Line 357 – dysfunctional |
Reply: |
Corrected |
15) |
Line 395 – fishing |
Reply: |
Corrected |
16) |
Line 519 – add in the EU finding under which this was implemented. |
Reply: |
This research received no external funding. |
Reviewer 3 Report
Title: The title looks like a status report for the recreational fisheries in Greece. A link between these information with the sustainability of either fishery resource or the fishery industry is recommended.
Abstract
L21: Please identify or clarify what’s ecological importance refers to here.
INTRODUCTION
L62: spelling error found. It should be “resources” instead of “recourses”
L65-69: definitions regarding recreational, commercial, semi-professional, and semi-recreational fishers need to be clarified. These may affect the population size being surveyed. Also, clarify “why the conflict between recreational and commercial fishers creating another significant identification issue for small-scale fisheries in Greece?”
L104-119: some kind of confusion. Telephone survey or interview? It should be the telephone survey for this study. Keep consistency throughout the text for the terms used.
In addition, as indicated that a nationwide telephone survey was conducted twice, the differences between these two need to be identified. Which part and/or how the precision was improved in the survey conducted in 2019? Also, a sample of 5,500 interviews was repeated? This number is different from those described in the Materials and Methods (M&M) section? Please clarify.
Materials and methods
L127-130: as indicated above, how the recreational, commercial, semi-professional, and semi-recreational fishers were defined? Is the population of MRF defined here include or partially overlap with those individuals defined above? If this was the case, the number of MRF estimated in the study would be overestimated. Please clarify.
L133-145: Spatially stratified sampling design presented here needs to be clarified. Why did some areas need to be oversampled? Why did additional weightings of the sample need to be applied for some groups and how? Please clarify, as these may have significant impacts on the results presented in this study.
L152-154: for the households survey where there was more than one member of the household engaged in recreational fishing, this information was recorded but it was not included in the final estimation. Please clarify which part of the estimations and why?
L157: The methods of fishing, avidity, and expenses need to be clearly specified.
L164-170: as indicated above, the raised total will be affected by the sampling design indicated in L133-145. It deserves authors to provide more detailed information on why it is necessary for weighting and oversampling, and how?
L174-175: Why a significance level of 0.01% instead of 0.05% was used? Are there any specific reasons?
L183: Use past tense in the results. Shall it be “was estimated to be 730,514”?
L204-205: Figure 2. The origin of the axis should be the “zero”. Also, add an axis break between the origin and the tick mark of 15-25. There were also overlaps found for the tick labels for the X-axis, it should be 15-24, 25-34, 35-44 … etc.
L211-212: Unnecessary decorations in Figure 3? This is a box plot. Why different shapes of the plot were used? Are there any special meanings?
L213-227: This is an invalid analysis. I don’t think the fishing pressure can be represented as the number of MRF in an area, because fishers who live in one area do not necessarily also fished in the same area, and the number of MRF may not be directly correlated with the total catch of the region. As indicated in L225, 30.80% of MRF travel further distances for fishing (neither to home RU nor to a neighboring RU). The catch rate and the frequency of fishing should be more related. Why not use these data? Thus, the result shown in Figure 4 is misleading.
L232-240: Apparently, there were some overlaps among different fishing modes, how these were counted in Figure 3? Why do all numbers in Figure 5 add up to 97.79%? If there were some overlaps, it should be more than 100%, isn’t it?
L246-250: The methods of fishing presented here also need to be described in the M&M section.
L281-288: The Y-axis in Figure 6 should start with “zero”. The estimation of the annual catch for RF was based on the number of MRF which will be affected by the sampling design indicated above, and need to be carefully checked.
L306-322: The annual expenses for MRF need to be specified in the M&M section. Which kind of items has been included? The absolute value may not be comparable among countries as the national income for each country is different.
L323-325: This is the M&M, not the results.
L330-349: The data presented here need to be carefully checked. If the number of MRF was estimated to be 730,514 (L183), then, the 11.04% would be ~3,356 instead of 90,000. Which one is correct? Please clarify.
L347-349: I could not understand what authors intended to say for “significant differences (p<0.01) can be detected amongst MRF aged between 25-34 years and unemployed who are in favor of better training, while married fishermen are in favor of stricter controls. What purposes of these comparisons?
DISCUSSION
Overall, the discussion presented in the manuscript is lengthy and redundant. There is no need to discuss every point described in the results. Only major and significant findings and how these are related to the fishery industries or economics as well as the management of fishery resources in Greece need to be focused on and discussed.
L352-392: Previous estimations although may not be as accurate as of the current study, it is also necessary to realize the potential errors that might be occurred in this study due to sampling design or over-counting … as indicated above.
L394-411: these discussions seem meaningless as no. of boats and no. of MRF is clearly not related.
L412-419: what are the purposes for discussing the participation rate of women? The possible reasons provided by authors also were subjective and lack of literature support.
L420-447: the discussion on the relationships between age and fishing mode, traveling for fishing, and avidity of fishers seem to provide no insight on the sustainability of either fishery industry or resource in the region. The possible reasons discussed for the observed patterns also were kind of subjective and lack of any literature support. I suggest authors either remove or condense these discussions, and focus on how these observations are important in terms of the management of recreational fisheries in Greece.
L448-473: The discussion provided here seems beyond the scope of this study and kind of misleading. Before fish farms are established, the original ecosystem could have already been dominated by the Sparidae family as shown in Figure 7. Fish farming for the Sparidae may indeed push this dominance further to some extent, but Figure 7 as it stands, could not tell you which one will be the case. Besides, whether escaped fish from fish farms could recruit to, and contribute an overall abundance of these species in the region or not were not known. The potential threat to natural biodiversity and the socioeconomic impacts also were not evaluated in the study and should not be included as the major results indicated in the abstract.
L474-485: The expenditure of MRF among countries may not be comparable as the per capita income for each country is different. Expenditure relative to the per capita income may be more appropriated. Also, how the 132 € million in L484 was estimated? If each individual spends 180.95€ per year, a total of 30,514 (L183) MRF would add up to 5.5 million? Please clarify.
L500: I don’t see any ecological importance that has been evaluated in this study. The only possible impact was mentioned in L472, which was inferred from other studies as well and was not the result of this study.
Author Response
Dear sir/madam
Thank you for carefully reading our manuscript and offering your comments to improve it. We have taken them into consideration in preparing the revised form of manuscript. Below is our point-by-point response to your comments. The appropriate corrections to the revised manuscript are presented with track changes as suggested. We hope that our interventions based on your comments increased the quality of our work and render our manuscript suitable for publication.
Yours Sincerely
Anastasios Papadopoulos
General comment from the authors: Marine recreational fishing is an understudied activity in most parts of the world, certainly in the Mediterranean. Since the present study is one of the first in the Eastern Mediterranean, covering all modes and methods of marine recreational fishing at a national level, it is, in our view, useful to provide detailed results of social or economic interest and attempt to interpret them. These results could be used as a reference point for future comparisons.
1) |
Title: The title looks like a status report for the recreational fisheries in Greece. A link between these information with the sustainability of either fishery resource or the fishery industry is recommended. |
Reply: |
We agree, thank you for the suggestion. The title has changed to: ‘‘Evaluation of marine recreational fisheries and their relation to sustainability of fisheries resources in Greece’. |
2) |
Abstract L21: Please identify or clarify what’s ecological importance refers to here |
Reply: |
We agree, as also stated in your comment in the discussion section “I don’t see any ecological importance that has been evaluated in this study”, we have deleted the ecological term in the abstract. |
3) |
INTRODUCTION L62: spelling error found. It should be “resources” instead of “recourses” |
Reply: |
Corrected. |
4) |
L65-69: definitions regarding recreational, commercial, semi-professional, and semi-recreational fishers need to be clarified. These may affect the population size being surveyed. Also, clarify “why the conflict between recreational and commercial fishers creating another significant identification issue for small-scale fisheries in Greece?” |
Reply: |
The line was rephrased avoiding semi-professional, and semi-recreational fishers See page 2. The word conflict was removed |
5) |
L104-119: some kind of confusion. Telephone survey or interview? It should be the telephone survey for this study. Keep consistency throughout the text for the terms used |
Reply: |
The methodology is explained within L121-L126. However the description is improved with the addition of a more detailed paragraph and the phrase telephone interview where appropriate. |
6) |
In addition, as indicated that a nationwide telephone survey was conducted twice, the differences between these two need to be identified. Which part and/or how the precision was improved in the survey conducted in 2019? Also, a sample of 5,500 interviews was repeated? This number is different from those described in the Materials and Methods (M&M) section? Please clarify |
Reply: |
Clarification added in L110. The aim of the current study is to present the main outcomes of the 2019 nation-wide telephone survey and not to compare the two surveys. The first survey with a sample of 5,500 interviews was conducted for standardizing our methodology and identifying possible weaknesses. The same sample of the 5,500 interviews was not repeated in 2019. A different sample was used. With the second survey we managed to make more accurate inferences about that large and heterogeneous population such as recreational fishers than for a tiny and homogeneous one, and therefore a higher number of units was needed than for coarser estimates. See also point 8. |
7) |
L127-130: as indicated above, how the recreational, commercial, semi-professional, and semi-recreational fishers were defined? Is the population of MRF defined here include or partially overlap with those individuals defined above? If this was the case, the number of MRF estimated in the study would be overestimated. Please clarify. |
Reply: |
At the start of the survey, the interviewer asks the interviewee if he or she had been engaged at least once in recreational fishing for the past 12 months prior to the survey. A professional fisher fishing for pleasure, is a recreational fisher, and no one knows if the fish will be illegally sold. We added however more detailed definitions in order to clarify the different terms. See also reply to point 4. |
8) |
L133-145: Spatially stratified sampling design presented here needs to be clarified. Why did some areas need to be oversampled? Why did additional weightings of the sample need to be applied for some groups and how? Please clarify, as these may have significant impacts on the results presented in this study. |
Reply: |
The reason for oversampling was to increase the possibility of MRF participation. Therefore, weights were applied in order to make the sample representative of the population in terms of geography and basic demographic groups. However additional information was added in the manuscript. |
9) |
L152-154: for the households survey where there was more than one member of the household engaged in recreational fishing, this information was recorded but it was not included in the final estimation. Please clarify which part of the estimations and why? |
Reply: |
The survey was conducted on a random sample of respondents, by phone calls to both landline and mobile phone numbers. There was a question about other members in the respondent’s household (apart from the respondent) being MRFs but if the answer was affirmative, there was no subsequent question to record the actual number of MRFs in that particular household. Therefore, the estimations are done at the respondent level. |
10) |
L157: The methods of fishing, avidity, and expenses need to be clearly specified. |
Reply: |
Done. Description of the methods were added. |
11) |
L164-170: as indicated above, the raised total will be affected by the sampling design indicated in L133-145. It deserves authors to provide more detailed information on why it is necessary for weighting and oversampling, and how? |
Reply: |
The oversampling was necessary in order to increase the probability of finding MRFs for the purposes of the survey, while the weighting was necessary in order to provide results that are representative to population distributions. See also reply to point 8. |
12) |
L174-175: Why a significance level of 0.01% instead of 0.05% was used? Are there any specific reasons? |
Reply: |
Due to the large sample size, we have chosen the significance level of 0.01% instead of 0.05% in order to be more accurate/strict in our interpretation of statistically significant differences between socio-demographic groups. |
13) |
L183: Use past tense in the results. Shall it be “was estimated to be 730,514”? |
Reply: |
Corrected |
14) |
L204-205: Figure 2. The origin of the axis should be the “zero”. Also, add an axis break between the origin and the tick mark of 15-25. There were also overlaps found for the tick labels for the X-axis, it should be 15-24, 25-34, 35-44 … etc. |
Reply: |
Corrected on figure 2. |
15) |
L211-212: Unnecessary decorations in Figure 3? This is a box plot. Why different shapes of the plot were used? Are there any special meanings? |
Reply: |
The “violin” plots (colored shapes around the boxplots) visualize the probability density of the sample at different values. We believe that this visual representation promptly conveys useful information on data distribution to the user. However, we acknowledge that some additional information on these plots should be added to the figure 3 legend. Please, see figure 3 legend. |
16) |
L213-227: This is an invalid analysis. I don’t think the fishing pressure can be represented as the number of MRF in an area, because fishers who live in one area do not necessarily also fished in the same area, and the number of MRF may not be directly correlated with the total catch of the region. As indicated in L225, 30.80% of MRF travel further distances for fishing (neither to home RU nor to a neighboring RU). The catch rate and the frequency of fishing should be more related. Why not use these data? Thus, the result shown in Figure 4 is misleading. |
Reply: |
Fishing pressure described in these lines is based on the RU where the respondents fish but it is not their residence. |
17) |
L232-240: Apparently, there were some overlaps among different fishing modes, how these were counted in Figure 3? Why do all numbers in Figure 5 add up to 97.79%? If there were some overlaps, it should be more than 100%, isn’t it? |
Reply: |
Every interviewee could select more than one modes of fishing, because MRFs could fish in various ways in one year even within the same day. Some respondents also added unknown modes of fishing which were defined as “other” It is explained in the relevant paragraph were the missing 2,21% is presented. |
18) |
L246-250: The methods of fishing presented here also need to be described in the M&M section. |
Reply: |
Done. Description is added M&M section. |
19) |
L281-288: The Y-axis in Figure 6 should start with “zero”. The estimation of the annual catch for RF was based on the number of MRF which will be affected by the sampling design indicated above, and need to be carefully checked. |
Reply: |
Corrected on figure 6. |
20) |
L306-322: The annual expenses for MRF need to be specified in the M&M section. Which kind of items has been included? The absolute value may not be comparable among countries as the national income for each country is different. |
Reply: |
The question was asking about any possible expenditure related to RF without mentioning specific items. The annual expenses are the total expenditure per fisher related to the activity during the past 12 months prior to the survey. It is not within the scope of this study to examine the relationship between expenditure and GDP for this or other countries. See also answer in point 29. |
21) |
L323-325: This is the M&M, not the results. |
Reply: |
We agree. The sentence was transferred to the M&M |
22) |
L330-349: The data presented here need to be carefully checked. If the number of MRF was estimated to be 730,514 (L183), then, the 11.04% would be ~3,356 instead of 90,000. Which one is correct? Please clarify. |
Reply: |
The 11,04% of 730,514 is 80,648. The 2% of 730,516 is 14.610. Added together (11,04 and 2%) equals 13,04% = 95,258 fishers (>90,000) that they are not aware of the rules and limits governing the activity they are partaking in. |
23) |
L347-349: I could not understand what authors intended to say for “significant differences (p<0.01) can be detected amongst MRF aged between 25-34 years and unemployed who are in favor of better training, while married fishermen are in favor of stricter controls. What purposes of these comparisons? |
Reply: |
It is not a comparison. These are reported results arising from the analysis. The interesting information is that the younger ages who are less aware of the rules and limits governing the activity are in favor of better training. That clarification is added in the relevant paragraph in the discussion. |
24) |
Overall, the discussion presented in the manuscript is lengthy and redundant. There is no need to discuss every point described in the results. Only major and significant findings and how these are related to the fishery industries or economics as well as the management of fishery resources in Greece need to be focused on and discussed. L352-392: Previous estimations although may not be as accurate as of the current study, it is also necessary to realize the potential errors that might be occurred in this study due to sampling design or over-counting … as indicated above. |
Reply: |
We agree with the comment. Τhe discussion in many cases is lengthy and redundant. The discussion is downsized (especially the proposed lines), several changes were made, comments deleted, clarifications were added, that can be traced through the track changes in the revised manuscript. Please see also answers to comments 8, 9, 11, 12 regarding sampling design. |
25) |
L394-411: these discussions seem meaningless as no. of boats and no. of MRF is clearly not related. |
Reply: |
We agree. Paragraph deleted. |
26) |
L412-419: what are the purposes for discussing the participation rate of women? The possible reasons provided by authors also were subjective and lack of literature support. |
Reply: |
Studies from other countries (see references in lines L415-416) report few women involved in recreational fishing, indicating that the activity is considered a predominantly male activity in Mediterranean countries. We believe that the increased participation rate of women in Greece is an issue of importance. Increased number of women fishers leads to increased number of fishers in the sea and an increased number of percentage of fishers compared to other countries. Additionally the avidity, modes and methods used by women can also have proportionally a small or large impact to the stocks. This study revealed that women use mainly lines from the shore (L248) a less effective than others, method of fishing. We believe that the involvement of women in recreational fishing should be further investigated. There is no known literature supporting the reasons for the low participation of women in recreational fishing. However relevant adjustments were made in the noted paragraph. |
27) |
L420-447: the discussion on the relationships between age and fishing mode, traveling for fishing, and avidity of fishers seem to provide no insight on the sustainability of either fishery industry or resource in the region. The possible reasons discussed for the observed patterns also were kind of subjective and lack of any literature support. I suggest authors either remove or condense these discussions, and focus on how these observations are important in terms of the management of recreational fisheries in Greece. |
Reply: |
The relationships between the above mentioned parameters certainly affects MRF activity and in this way can directly affect the quantities and the species οf the removed fish and thus the sustainability of the ecosystem. However, their importance and relationship is not assessed in this study but it should be a subject of future research. Thus, the relevant part of the discussion is rephrased, bibliography is added and the discussion condensed. See also reply to comment 24. |
28) |
L448-473: The discussion provided here seems beyond the scope of this study and kind of misleading. Before fish farms are established, the original ecosystem could have already been dominated by the Sparidae family as shown in Figure 7. Fish farming for the Sparidae may indeed push this dominance further to some extent, but Figure 7 as it stands, could not tell you which one will be the case. Besides, whether escaped fish from fish farms could recruit to, and contribute an overall abundance of these species in the region or not were not known. The potential threat to natural biodiversity and the socioeconomic impacts also were not evaluated in the study and should not be included as the major results indicated in the abstract. |
Reply: |
It might be possible that the original ecosystem could have already been dominated by the Sparidae family. However figure 7 shows the present MRF catches as reported by the MRF and is not referring to a past period nor a comparison between the present and the past is attempted in this study. Also in this paper we examine the relationship between fish farms and Sparus aurata solely not the sparidae family in general. There is strong evidence supported by bibliography that reared seabreams contribute substantially to the natural population in many ways and with direct ecological implications. What is examined in this work is the relationship between recreational fishing activity and the particular species and gives an explanation why the most caught species by MRF is Sparus aurata. |
29) |
L474-485: The expenditure of MRF among countries may not be comparable as the per capita income for each country is different. Expenditure relative to the per capita income may be more appropriated. Also, how the 132 € million in L484 was estimated? If each individual spends 180.95€ per year, a total of 30,514 (L183) MRF would add up to 5.5 million? Please clarify. |
Reply: |
We agree that expenditure is a complicated issue to deal with. Although Hyder 2018, reports that there is no significant correlation between per capita GDP and expenditure for the Mediterranean, we have decided to keep only the average annual expenses of the European MRF and downsize the discussion on expenditure, as it is not within the scope of this study to examine this matter in depth. The number in L183 is 730,514 not 30,514. |
30) |
L500: I don’t see any ecological importance that has been evaluated in this study. The only possible impact was mentioned in L472, which was inferred from other studies as well and was not the result of this study. |
Reply: |
We agree. Ecological importance was removed |
Reviewer 4 Report
The paper needs some corrections. I put its in text.

Author Response
Dear sir/madam
Thank you for carefully reading our manuscript and offering your comments to improve it. We have taken them into consideration in preparing the revised form of manuscript. Below is our point-by-point response to your comments. The appropriate corrections to the revised manuscript are presented with track changes as suggested. We hope that our interventions based on your comments increased the quality of our work and render our manuscript suitable for publication.
Yours Sincerely
Anastasios Papadopoulos
Comments and Suggestions for Authors
The paper needs some corrections. I put its in text.
1) |
Please, put the whole word, without a hyphen, i.e. approximately. It is available for all text. |
Reply: |
Unfortunately, this format is obligatory by the template of the sustainability journal. Even if one word changes, another one will have a hyphen. |
2) |
Please put the references in text according Instructions for Authors. |
Reply: |
References placed in text according to Instructions for Authors. |
3) |
modes. |
Reply: |
Corrected |
4) |
The Conclusions chapter is missing. You can put this paragraph. You can structure it according Instructions for Authors. |
Reply: |
Corrected |
Round 2
Reviewer 1 Report
- The format of the section title of discussion needs correction.
- The 1st two paragraphs of the discussion may be combined and revised.
Author Response
Dear sir/madam
Thank you for carefully reading our manuscript for the second time and offering your comments to improve it. We have taken them into consideration in preparing the revised form of manuscript. Below is our point-by-point response to your comments. The appropriate corrections to the revised manuscript are presented with track changes as suggested. We hope that our interventions based on your comments increased the quality of our work and render our manuscript suitable for publication.
Yours Sincerely
Anastasios Papadopoulos
1. |
The format of the section title of discussion needs correction. |
Reply: |
The format of the section title of discussion corrected. |
2. |
The 1st two paragraphs of the discussion may be combined and revised. |
Reply: |
The first two paragraphs of the discussion were combined and revised. |
Reviewer 2 Report
The revision has been undertaken in detail by the authors, and in my view the manuscript is now ready to proceed to publication.
Author Response
Dear sir/madam
Thank you for carefully reading our manuscript for the second time.
Yours Sincerely
Anastasios Papadopoulos
Reviewer 3 Report
Most of my comments have been addressed although a few of them (such as replies 16, 20, and 24) are still not quite clearly explained. The paper is now in better shape, and suitable for publication in the journal.
Author Response
Dear sir/madam
Thank you for carefully reading our manuscript for the second time and offering your comments to improve it. We have taken them into consideration in preparing the revised form of manuscript. Below is our point-by-point response to your comments. The appropriate corrections to the revised manuscript are presented with track changes as suggested. We hope that our interventions based on your comments increased the quality of our work and render our manuscript suitable for publication.
Yours Sincerely
Anastasios Papadopoulos
---
Comments and Suggestions for Authors
Most of my comments have been addressed although a few of them (such as replies 16, 20, and 24) are still not quite clearly explained. The paper is now in better shape, and suitable for publication in the journal.
16) |
L213-227: This is an invalid analysis. I don’t think the fishing pressure can be represented as the number of MRF in an area, because fishers who live in one area do not necessarily also fished in the same area, and the number of MRF may not be directly correlated with the total catch of the region. As indicated in L225, 30.80% of MRF travel further distances for fishing (neither to home RU nor to a neighboring RU). The catch rate and the frequency of fishing should be more related. Why not use these data? Thus, the result shown in Figure 4 is misleading. |
Reply |
The questionnaire used in our survey, apart from recording the residence prefecture of the MRFs, included a question asking the prefectures where MRFs actually fish. This map has been created based on the latter, the prefecture where MRFs fish. With this analysis we were aiming to provide an indication of the spatial distribution of MRF fishing pressure within the country. We agree with the reviewer that this mapping would be more informative if it were combined with MFRs catches. And it is actually a work under preparation right now. |
20) |
L306-322: The annual expenses for MRF need to be specified in the M&M section.Which kind of items has been included? The absolute value may not be comparable among countries as the national income for each country is different. |
Reply: |
The annual expenses for MRF that were requested during the survey are now specified in the M&M section. Data on the costs recreational fishers incur help to explain their behavior and are useful in understanding the wider economic impact of this fishing activity. The question was asking about any possible expenditure related to RF without mentioning specific items since the use of a detailed list of the expenditure requires a survey of its own. The annual expenses are the total expenditure per fisher related to the activity during the past 12 months prior to the survey and included all expenditures within the recall period with the exception of fishing license fees, since such expenses do not exist in Greece. It is not within the scope of this study to examine the relationship between expenditure and GDP for this or other countries. |
24) |
Overall, the discussion presented in the manuscript is lengthy and redundant. There is no need to discuss every point described in the results. Only major and significant findings and how these are related to the fishery industries or economics as well as the management of fishery resources in Greece need to be focused on and discussed. L352-392: Previous estimations although may not be as accurate as of the current study, it is also necessary to realize the potential errors that might be occurred in this study due to sampling design or over-counting … as indicated above. |
Reply: |
The authors agreed with the comment and therefore the discussion is downsized focusing especially to the proposed lines and several changes were made, comments deleted, clarifications added, that can be traced through the track changes in the revised manuscript. Further changes were made and the remaining typos and syntax errors were corrected following the second revision. |
Reviewer 4 Report
Some corrections are needed. I put my suggestions in text. Please respect the Instructions for Authors of Sustainability Journal - https://www.mdpi.com/journal/sustainability/instructions#preparation

Author Response
Dear sir/madam
Thank you for carefully reading our manuscript for the second time and offering your comments to improve it. We have taken them into consideration in preparing the revised form of manuscript. Below is our point-by-point response to your comments. The appropriate corrections to the revised manuscript are presented with track changes as suggested. We hope that our interventions based on your comments increased the quality of our work and render our manuscript suitable for publication.
Yours Sincerely
Anastasios Papadopoulos
--
Comments and Suggestions for Authors
1. |
Some corrections are needed. I put my suggestions in text. Please respect the Instructions for Authors of Sustainability Journal - https://www.mdpi.com/journal/sustainability/instructions#preparation |
Reply: |
The format of the section title of discussion corrected and instructions for authors have been followed were necessary. |
Round 3
Reviewer 1 Report
The 1st two paragraphs in the Discussion need editing.
Reviewer 4 Report
I have no comment.